# Advancements in LAMP-Based Diagnostics: Emerging Techniques and Applications in Viral Detection with a Focus on Herpesviruses in Transplant Patient Management

**DOI:** 10.3390/ijms252111506

**Published:** 2024-10-26

**Authors:** Ana Cláudia Martins Braga Gomes Torres, Carolina Mathias, Suelen Cristina Soares Baal, Ana Flávia Kohler, Mylena Lemes Cunha, Lucas Blanes

**Affiliations:** 1Laboratory for Applied Science and Technology in Health, Carlos Chagas Institute, Oswaldo Cruz Foundation (Fiocruz), Curitiba 81350-010, Brazil; lucas.blanes@fiocruz.br; 2Post-Graduation Program in Genetics, Department of Genetics, Federal University of Parana, Curitiba 81530-980, Brazil; carolina.mathias@ufpr.br (C.M.); s.soares@ufpr.br (S.C.S.B.); anakohler@ufpr.br (A.F.K.); mylenacunha@ufpr.br (M.L.C.)

**Keywords:** point-of-care, LAMP, viral detection, Herpesviridae, isothermal amplification

## Abstract

Loop-mediated isothermal amplification (LAMP) is a highly effective molecular diagnostic technique, particularly advantageous for point-of-care (POC) settings. In recent years, LAMP has expanded to include various adaptations such as DARQ-LAMP, QUASR, FLOS-LAMP, displacement probes and molecular beacons. These methods enable multiplex detection of multiple targets in a single reaction, enhancing cost-effectiveness and diagnostic efficiency. Consequently, LAMP has gained significant traction in diagnosing diverse viruses, notably during the COVID-19 pandemic. However, its application for detecting Herpesviridae remains relatively unexplored. This group of viruses is of particular interest due to their latency and potential reactivation, crucial for immunocompromised patients, including organ and hematopoietic stem cell transplant recipients. This review highlights recent advancements in LAMP for virus diagnosis and explores current research trends and future prospects, emphasizing the detection challenges posed by Herpesviridae.

## 1. Introduction

Loop-mediated isothermal amplification (LAMP) is a nucleic acid amplification technique (NAAT) that offers numerous advantages, including being isothermal, fast, and low-cost. Among NAATs, the gold standard method is quantitative PCR (qPCR), known for its high sensitivity and specificity, making it highly effective for pathogen detection and quantification. However, qPCR has limitations, particularly due to its reliance on expensive thermal cycling equipment, trained personnel, and complex sample preparation. These requirements restrict its use in resource-limited environments and point-of-care (POC) diagnostics [1].

In comparison to other isothermal NAATs like NASBA (sequence-based amplification), RPA (recombinase polymerase amplification), and HDA (helicase-dependent amplification), LAMP stands out due to its simplicity and versatility. NASBA, while eliminating the need for thermal cycling, is more technically demanding and less flexible in assay design. RPA and HDA, though also isothermal, require specialized reagents or equipment, such as recombinase proteins or helicase, which adds complexity [2]. LAMP, by contrast, operates at a constant temperature, requires fewer reagents, and allows for visual detection (color change), making it particularly suitable for resource-limited settings and point-of-care diagnostics. Furthermore, LAMP’s tolerance to inhibitors in clinical samples reduces the need for extensive sample preparation, offering a significant advantage over these alternative techniques. Additionally, LAMP offers the possibility of detecting DNA amplification through a color change, eliminating the need for sophisticated detection systems such as lasers or fluorescence readers. This visual detection method makes the reaction more accessible, especially in resource-limited settings, where advanced equipment may not be available. Moreover, LAMP offers rapid results, with reactions typically completed in 30–60 min, in contrast with qPCR that usually takes around 1 h and 30 min [1].

It has been applied to detect a wide range of microorganisms, such as viruses, bacteria, protozoa, and helminths [3]. Additionally, LAMP has been utilized in cancer research, such as detecting mutations and the translocations in chronic myeloid leukemia [4], as well as combining techniques such as lateral flow assays, gel electrophoresis, and mobile phone detection for the detection of HPV16 [5]. Moreover, LAMP has advanced the development of new instruments and technologies, including point-of-care (POC) lab-on-a-chip devices and other innovative tools for detecting SARS-CoV-19 [6,7] Additionally, LAMP also allows for multiplexing reactions, including the use of fluorophores for quantifying pathogen detection [8]. Due to the versatility of the technique, it is becoming evident that LAMP is an excellent choice for detecting pathogens, including Herpesvirus, the main focus of this article.

Herpesviridae is a unique group of viruses characterized by their ability to alternate between lytic and latent phases [9]. Once an individual is infected, the virus remains in the body for a long time due to its latency mechanisms. Herpesviruses have genomes ranging from 120 to 250 kilobases (kb). Their gene replication occurs in a specific order: immediate early genes (which encode regulatory proteins), early genes (which encode proteins for DNA replication), and late genes (which encode viral capsid proteins) [10]. This orchestrated replication process is key to the virus’s success in evading the immune system. Within this group, eight types are particularly relevant to human health: Cytomegalovirus (CMV), the Epstein–Barr Virus (EBV), Herpes Simplex Virus 1 (HSV-1), Herpes Simplex Virus 2 (HSV-2), Human Herpesvirus 6 (HHV-6), Human Herpesvirus 7 (HHV-7), Human Herpesvirus 8 (HHV-8) or Kaposi’s sarcoma-associated herpesvirus (KSHV), and Varicella Zoster Virus (VZV) [11].

Globally, herpesviruses pose a significant health burden, particularly in immunocompromised populations, such as those with HIV/AIDS, transplant recipients, and neonates. Cytomegalovirus (CMV) affects 50–80% of the global adult population [12], while the Epstein–Barr Virus (EBV) is implicated in several cancers, such as Burkitt lymphoma and nasopharyngeal carcinoma [13]. These viruses, alongside others like the Varicella Zoster Virus (VZV), can cause lifelong infections with potential reactivation, leading to serious complications. In high-risk populations such as hematopoietic stem cell transplantation (HSCT) patients, herpesvirus reactivation can result in severe outcomes, including encephalitis, pneumonitis, and graft failure [14]. This emphasizes the urgent need for improved, rapid, and accessible diagnostic tools to guide clinical decisions and reduce mortality.

Among these, CMV is particularly prevalent and significantly impacts HSCT, with the highest risk of reactivation occurring in the first 100 days post-transplant. Reactivation can lead to pneumonia, a condition with a mortality rate of over 50% [15]. Additionally, CMV reactivation is associated with an increased risk of graft failure, graft-versus-host disease (GVHD), and secondary bacterial and fungal infections [16]. EBV reactivation, though less frequent, is particularly concerning due to its association with post-transplant lymphoproliferative disorder (PTLD), affecting around 10–15% of children compared with 1–3% of adults. This makes pediatric patients a high-risk group for EBV reactivation [17]. Beyond PTLD, EBV reactivation can lead to complications such as encephalitis, pneumonitis, and hepatitis. HHV-6 reactivation presents severe complications, with encephalitis affecting around 11.6% of HSCT patients, alongside rash and fever, which contribute to the development of GVHD [18]. Therefore, monitoring viral load in these patients is critical to prevent mortality and graft loss.

The development of rapid, accurate, and accessible diagnostic methods for herpesviruses is urgently needed. However, a major limitation of LAMP compared with qPCR is the difficulty in determining an accurate viral load. To overcome these limitations, making modifications in the assay, such as fluorophore inclusion, is a possibility discussed in this review. In addition, this review aims to provide a comprehensive overview of LAMP and its potential applications for herpesvirus detection. LAMP provides a promising alternative for the molecular detection of herpesviruses due to its advantages and potential for including multiple targets to optimize detection. This review aims to explore LAMP’s potential applications for herpesvirus detection, encouraging further research in this area by highlighting its effectiveness and versatility.

## 2. LAMP Methodology Overview

LAMP is a nucleic acid amplification technique (NAAT) that amplifies a target sequence at a constant temperature. This advantage is due to the use of Bst polymerase, an enzyme isolated from *Bacillus stearothermophilus*, which has strand displacement activity. This enzyme can synthesize a new DNA strand while simultaneously displacing the original DNA strand, allowing for continuous DNA amplification without the need for heat-induced denaturation. This eliminates the need for thermal cycling, making the amplification process occur under isothermal conditions of around 60–65 °C.

The simplicity of performing the technique in basic equipment, such as a hot water bath or heating block, makes LAMP highly attractive compared with other NAATs [19]. The principle of LAMP was first described by Notomi et al. (2000), and several modifications have been proposed to improve the specificity and sensitivity of the technique [20]. Below are the LAMP steps, which involve primer design and amplification.

### 2.1. LAMP Primers

Initially, a set of four primers complementary to six regions within the target sequence was designed (Figure 1) [20,21]. Common tools for LAMP primer design include PrimerExplorer https://primerexplorer.jp/e/ (accessed on 10 October 2024) and the NEB^®^ LAMP Primer Design Tool: https://lamp.neb.com/#!/ (accessed on 10 October 2024). Careful primer design is essential to avoid non-specific amplification. Sequences must be analyzed using in silico tools, such as nucleotide BLAST (NCBI), to ensure target specificity.

The sequences in the target region are designated as F3c, F2c, and F1c at the 3′ end and B1, B2, and B3 at the 5′ end. LAMP primers consist of two inner primers, called the forward internal primer (FIP) and backward internal primer (BIP), and two outer primers, the forward outer primer (F3), complementary to F3c, and backward outer primer (B3). The FIP consists of F1c and F2, while the BIP contains B1c (complementary to B1) and B2. The inner primers, containing both sense and antisense sequences, anneal to form a loop structure, while the outer primers are responsible for strand displacement during the non-cyclic stage of LAMP [20]. Additionally, loop primers—loop primer forward (LPF) and loop primer backward (LPB)—anneal to the loop structures in the LAMP amplicons, increasing the number of starting points for amplification. The use of loop primers has been shown to significantly improve the efficiency and sensitivity of the reaction [21].

### 2.2. LAMP Reaction

At the beginning of the LAMP reaction, the FIP binds to the F2c region on the target strand and initiates the synthesis of a complementary DNA strand (Figure 1). The F3 outer primer attaches to the F3c region and starts strand displacement DNA synthesis, releasing the FIP-linked complementary strand and forming a loop structure at one end. Subsequently, the BIP and B3 primers also direct strand displacement DNA synthesis, creating another loop structure at the other end, resulting in a dumbbell-shaped structure with a stem-loop at each end of the target sequence. This dumbbell structure is then used as a template for further amplification cycles in self-primed DNA synthesis under isothermal conditions. Additional LPF and LPB primers anneal to the artificial template with the dumbbell structure, increasing the selectivity and efficiency of the reaction. The set of six primers, which are complementary to eight different regions of the target sequence, ensures high specificity and a rapid amplification rate in a short reaction time. The final products of the LAMP reaction are stem-loop DNAs and cauliflower-like structures with multiple loops [20,21].

### 2.3. LAMP Advantages

This method offers several advantages, making it optimal for various applications in research, such as biomonitoring and clinical diagnostics [22]. The use of specifically designed primers allows for the discrimination between closely related DNA sequences. However, all parameters must be optimized to reduce the likelihood of false-positive results, which can occur due to primer-dimer formation (when two primers bind to each other instead of the target DNA sequence) or carry-over contamination (contamination with amplified DNA from a previous reaction). Additionally, LAMP’s sensitivity is comparable to or even higher than that of the polymerase chain reaction (PCR), allowing for the detection of low concentrations of target DNA [1].

LAMP also exhibits less sensitivity to inhibitory substances present in complex biological samples, enabling direct analysis without the need for extensive sample pre-processing. This feature is particularly advantageous in situations requiring rapid, on-site testing, such as during infectious disease outbreaks, like the SARS-CoV-2 pandemic [23]. Although LAMP is often used as a pathogen-specific diagnostic tool, it can be easily adapted for multiplexing (mLAMP), allowing for the simultaneous detection of multiple targets within a single reaction [24,25,26,27].

One of the main advantages of the LAMP technique is the ability to visually detect amplification results, making it an accessible option for resource-limited settings. Conventional methods for detecting LAMP products include visualization in agarose gel electrophoresis, monitoring turbidity caused by the precipitation of magnesium pyrophosphate (a by-product of the reaction), using intercalating fluorescent dyes such as SYBR green, or employing pH-sensitive indicators [28,29] (Figure 2). Furthermore, LAMP can be combined with other molecular techniques to enhance the specificity of visual detection. Table 1 presents a series of examples of LAMP reactions based on these methodologies for detecting various viruses.

As demonstrated, LAMP has been applied to a wide range of targets. However, aspects such as the limit of detection (LoD), sensitivity, and specificity often require further investigation by researchers to optimize and improve the technique’s accuracy. In the following sections, we will explain the most commonly employed detection methods in LAMP reactions.

## 3. LAMP Detection Methodologies

### 3.1. Electrophoresis

One of the standard methods for detecting LAMP products is agarose gel electrophoresis. In this process, the negatively charged DNA fragments migrate towards the positive pole when subjected to an electric current through an agarose gel matrix. Due to the unique structure of LAMP products, which often form dumbbell-shaped and cauliflower-like structures, electrophoresis typically reveals a ladder pattern rather than discrete bands, indicating the presence of amplification products (Figure 3). This smearing effect occurs because LAMP generates multiple fragments of varying sizes, making it harder to distinguish specific target products solely through electrophoresis. While this technique is effective for confirming amplification and can help determine the presence or absence of pathogen infection, it has limitations. Agarose gel electrophoresis operates in an open system, which increases the risk of contamination from amplified products during post-amplification steps. Consequently, complementary methods, such as real-time fluorescence detection or turbidity measurements, are often employed for more precise quantification and confirmation of LAMP results [44].

### 3.2. Turbidity Method

Mori et al. developed a turbidity-based method for detecting amplicons in LAMP reactions, utilizing the precipitation of magnesium pyrophosphate, a by-product of DNA amplification. During the LAMP reaction, DNA polymerase amplifies the target DNA, releasing pyrophosphate ions, which combine with magnesium ions to form a visible precipitate, increasing the turbidity of the reaction mixture. This approach addresses the limitations of traditional LAMP assays, which rely on agarose gel electrophoresis for detection—methods that are prone to contamination and require additional post-amplification processing [45].

The authors also explored the use of calcein, a fluorescent dye that binds to magnesium ions. In the presence of magnesium, calcein’s fluorescence is quenched, offering an alternative for real-time detection of LAMP amplicons. This dual detection method enhances the reliability and versatility of LAMP assays, making them particularly useful in point-of-care and field settings, where rapid and accurate diagnostics are crucial. It is important to note that turbidity signals in LAMP assays arise from by-products of DNA amplification rather than specific primer reactions. This can complicate the detection of non-specific amplification, making it crucial to validate primer sets rigorously [45]. Tools such as PrimerExplorer require strict testing to ensure primer specificity and minimize non-specific reactions before use in LAMP assays.

### 3.3. Colorimetric Method

Colorimetric Loop-Mediated Isothermal Amplification Detection is a powerful and innovative molecular biology technique used for the rapid and sensitive detection of specific DNA or RNA sequences. The colorimetric approach adds an extra layer of convenience to LAMP assays by allowing visual detection of amplification results without the need for sophisticated instruments, making it particularly well-suited for on-site molecular testing, especially in areas with limited resources [46].

In the conventional approach to colorimetric LAMP, indicators such as phenol red or hydroxynaphthol blue (HNB) are employed to induce a visible change in color (Figure 4) [47,48]. In LAMP reactions, the color change using indicators like HNB and phenol red is based on the pH shift caused by DNA amplification. Initially, HNB is blue under basic conditions, and as the reaction progresses and pyrophosphate ions accumulate, the pH decreases. This pH drop causes HNB to change from blue to colorless or yellow, indicating positive amplification. Similarly, phenol red starts as red and turns yellow as pH decreases during DNA amplification, providing a visual indicator of successful target amplification in LAMP assays.

Among the colorimetric approaches for virus detection, the LAMP method stands out as a preferred choice due to its rapid detection, high efficiency, cost-effectiveness, and reliable production of highly reproducible assay results [49,50]. The use of colorimetric LAMP in the diagnosis of viruses that affect humans and animals is quite broad. This methodology has been extensively investigated in the context of detecting the SARS-CoV-2 virus. Park et al. [51] demonstrated for the first time the effectiveness of using colorimetric detection based on RT-LAMP. Reverse Transcription Loop-Mediated Isothermal Amplification (RT-LAMP) combines reverse transcription (RT) with LAMP for the rapid and sensitive detection of RNA targets, making it particularly useful for the identification of RNA viruses or mRNA transcripts. This methodology was extensively used, for example, in the detection of SARS-CoV-2 from saliva. The protocol enabled rapid and sensitive detection of <10^2^ viral genomes per reaction in saliva samples [52]. In the literature, several articles describe the effectiveness and ability to detect SARS-CoV-2 using colorimetric methods [52,53,54,55,56,57].

More recently, Akter et al. [58] discussed the applicability of colorimetric LAMP in the detection of SARS-CoV-2 in untreated wastewater, supporting the use of RT-LAMP as a specific and efficient method for screening. This methodology was also capable of detecting SARS-CoV-2 from nasopharyngeal swab samples without RNA isolation [59].

Drawing from the evidence of numerous successful studies, such as the ones cited above, it becomes evident that colorimetric LAMP holds substantial methodological promise for detecting SARS-CoV-2 across various sample types. Moreover, this approach has been extensively applied for detecting other viral types, as further discussed below.

The influenza virus, commonly known as the flu, is a highly contagious viral infection that primarily affects the respiratory system. The influenza virus belongs to the Orthomyxoviridae family and is classified into types A, B, and C. Among these, influenza A and B are the main culprits for seasonal outbreaks and epidemics [60]. Due to its great infectious potential, several diagnostic methodologies have been developed to enable increasingly accurate, rapid, and cost-effective diagnoses, and colorimetric LAMP stands out as one of the used ones. Filaire et al. [61] recently developed a colorimetric LAMP (using a WarmStart Colorimetric LAMP 2X Master Mix kit (M1800, NEB, Hitchin, UK)) to detect highly pathogenic avian influenza viruses (HPAIV), and RT-LAMP assay allowed the specific detection of HPAIV H5Nx clade 2.3.4.4b within 30 min with a sensitivity of 86.11%. In the same way [62,63,64], other authors applied the same methodology, with some practical differences, to detect the avian influenza virus. In all of these studies, the methodology appears to be fast, sensitive, and efficient for detecting the virus.

Following the same logic, colorimetric LAMP was applied in a very satisfactory way in the detection of viruses that infect humans, animals, and plants, such as canine parainfluenza virus 5 (using hydroxynaphthol blue) [65], banana bract mosaic virus (using hydroxynaphthol blue) [66], Crimean-Congo hemorrhagic fever (using phenol red) [67], Apple mosaic virus (ApMV) and Prunus necrotic ringspot virus (PNRSV) (using hydroxynaphthol blue) [68], and Rift Valley fever virus (RVFV) (using phenol red) [69].

### 3.4. Fluorogenic Probe Method

Several strategies and possibilities in the detection of LAMP have been developed recently, including the use of fluorescent probes, similar to qPCR detection making possible, for example, viral load quantification (Table 1). In a LAMP fluorescence-based assay, a specific fluorescent probe is designed to bind to the amplified target sequence. In general, the probe consists of a fluorophore and a quencher molecule in close proximity, resulting in a quenching of the fluorescence signal [70]. During the LAMP process, the Bst DNA polymerase synthesizes new DNA strands, displacing the original strands and releasing the fluorophore and quencher, resulting in an increase in fluorescence signal. In addition, a strategy using 3′ exonuclease activity of Taq Polymerase together with Bst can be useful for fluorophore cleavage and signal detection [71]. Several fluorescence-based strategies have emerged in the context of LAMP, utilizing signal amplification or guanine dequenching mechanisms.

Amplification efficiency in fluorescence-based assays is measured by time-to-threshold (Tt), which is analogous to Ct in qPCR and reflects the time taken for the amplification’s fluorescence signal to surpass a predefined threshold and become detectable. The number of viral copies can be determined by generating standard curves, typically through 10-fold serial dilution using plasmid constructs, correlating the number of viral copies per microliter (copies/µL). Another method, the Plaque Formation Unit (PFU), involves infecting host cells with the virus and counting the plaques formed per unit volume. PFU/mL can then be related to Tt values to calculate viral load [72].

In LAMP assays, the coefficient of determination (R^2^) is used to assess the accuracy of viral quantification. By plotting Tt against viral copy numbers on a logarithmic scale (based on serial dilution), R^2^ reflects how well the Tt values align with the actual viral load. A high R^2^ value (close to 1) indicates a strong correlation, demonstrating reliable quantification. Alternatively, Relative Fluorescence Units (RFUs) can quantify viral load by measuring the fluorescence emitted during amplification. The RFU reflects the amount of DNA in the sample relative to a no-template control (NTC), with higher DNA concentrations yielding higher RFU values [73]. Intercalating dyes, such as SYBR Green^®^ or fluorescent probes, are often used to generate fluorescence, facilitating the calculation of linear regression and amplification efficiency. All these parameters help reach the limit of detection (LoD), which is defined as the lowest concentration of a target (such as viral DNA or RNA) that can be reliably detected by an assay.

In viral quantification using LAMP, both the LoD and assay sensitivity vary significantly based on factors such as the sample’s chemical composition, reaction inhibitors, and differences in amplification efficiency. For instance, Yaren et al. (2021), in a study on SARS-CoV-2 detection using a displacement probe, demonstrated varying LoD across different sample types: 200 copies/μL in saliva with 100% efficiency, 100 copies/μL in saliva collected on Q-paper (a specialized paper for preserving biological samples) with 40% efficiency, and 1000 copies/μL in nasal swabs with 100% efficiency [31].

Singleplex or multiplex reactions often suffer from reduced sensitivity due to primer competition and non-specific interactions within the reaction mix. This reduction in efficiency is more pronounced when using higher primer concentrations. For example, Yaren et al. (2021) observed that in multiplex assays, the RNAse P gene exhibited delayed amplification when high amounts of RNA were present [31]. In such cases, reaction optimization (e.g., adjusting primer concentrations, dNTPs, and magnesium ion levels) is crucial for maintaining high sensitivity and specificity.

The inclusion of multiple primers and probes can create competition for reaction reagents, impairing assay performance. Therefore, careful primer design is essential to ensure correct target amplification while avoiding non-specific amplification. Furthermore, assays should be designed to target conserved regions of the viral genome to ensure accurate detection. Regions prone to mutations or genes with highly repetitive sequences should be avoided. In the CMV genome, for instance, genes such as UL144, UL146, and UL147 show considerable variation and should be excluded from assay design [74]. The following sections will explore strategies for LAMP fluorescence-based assays, including the use of fluorescent probes to enhance detection specificity. These methods allow real-time monitoring of amplification, further improving sensitivity and reliability in viral quantification.

### 3.5. Assimilation Probe (Displacement Probe) Method

This technique was first described by Kubota in 2011 [75]. In this approach, loop primers (either loop primer backward (LPB) or loop primer forward (LPF)) are modified with a universal sequence—a standard sequence that is identical across all probes included in the assay—known as the F strand at the 5′ end, while a fluorophore is incorporated at the 3′ end. Additionally, an oligonucleotide complementary to the F strand, called the quencher strand, is included with a quencher molecule at its 3′ end (Figure 5). Initially, the probe anneals to the F strand, causing fluorescence quenching. As amplification proceeds, the newly synthesized DNA displaces the quencher strand, releasing the fluorophore and thus leading to fluorescence [71]. This approach is commonly referred to as strand displacement probes.

This method has been used to identify arboviruses like Zika (ZKV), Dengue (DENV), and chikungunya Virus (CHIKV), which are transmitted by the *Aedes aegypti* mosquito and are more prevalent in tropical countries [30]. Multiplex reactions were performed, labeling LPB at the 5′ end with FAM, TAMRA, and HEX, respectively. Since the quencher was complementary to the universal sequence included in all probes, a single quencher was sufficient. The LoD was 0.71 PFU for ZKV, 1.22 for DENV, and 37.8 copies for CHIKV.

In another study by the same group, Yaren et al. targeted the N, S genes of SARS-CoV-2 as well as the human RNase P gene [31]. The N and S genes were labeled with FAM, while the RNase P gene was labeled with JOE at the 3′ end of the probe, and LPF was labeled with the Iowa Black Darq Quencher (IABkFQ). The assay achieved a LoD of 25 copies of the N gene within 12 min, 10 copies of the S gene within 16 min, and 44 copies of the RNase P gene within 16 min [31], using virus isolate RNA from SARS-CoV-2. While the multiplex assay effectively targeted both SARS-CoV-2 and RNase P, a delay in detecting RNase P was noted, likely due to competition among LAMP reagents. This suggests that multiplexing LAMP assays should be approached cautiously, especially considering the higher copy numbers required for reliable detection.

Further work by Jang et al. applied assimilation probes (displacement probes) to detect SARS-CoV-2, targeting the RdRP, E, and N genes, with actin beta as an internal control (IC) [24]. LPB was labeled with FAM (RdRP), HEX (E), and Cy5 (N and IC), while quencher probes were attached to BHQ1 and BHQ2. The lowest LoD was achieved with a combination of the RdRP and N genes, detecting 10⁻⁵ to 10⁻⁶ viral copies based on a 10-fold serial dilution of clinical samples. No IC signal was detected at these low concentrations. However, IC signal was detected at 10⁻^2^ diluted samples from non-infected patients. Sensitivity of RT-LAMP was comparable to RT-qPCR: RdRP (93.85%), N (94.62%), and RdRP/N (96.92%) compared with AllplexTM 2019-nCoV (Seegene) (100%) [24].

More recently, Kline et al. developed an assimilation probe assay to detect three specific targets within the N gene of SARS-CoV-2 (Delta, Omega, and Omicron variants), including an internal control [32]. The study employed a thermostable chimeric polymerase, which is known for its enhanced tolerance to reaction inhibitors. The assay successfully detected all three targets across most strains, except for one target in the Omicron variant. The assay achieved an overall sensitivity of 87%, with 100% sensitivity in samples containing more than 30 viral copies.

Assimilation probes have also been applied in detecting foot-and-mouth disease virus (FMDV) with an LoD of 100 viral copies [33], porcine circovirus 3 (PCV3) with an LoD of 50 copies [34], and porcine epidemic diarrhea virus (PEDV) with an LoD of 10 copies/µL, including an internal control for the Sus scrofa β-actin gene [35]. This approach has also been successfully used for the detection of *Salmonella enterica* [76,77].

### 3.6. Detection of Amplification by Releasing of Quenching (DARQ-LAMP)

First described by Tanner et al. [78], the DARQ-LAMP technique involves a modified FIP, labeled with a quencher at the 5′ end in the Fc portion. Additionally, the approach includes a probe with a fluorophore at the 3′ end that is complementary to the Fc region of the FIP (Fd). Initially, Fd anneals to the F1c region, and this strand is released by F3 primer activity. The new strand is then annealed by the BIP through complementarity to the B2 region. The strand synthesized from the B2 target displaces Fd, releasing the probe and resulting in a gain of fluorescence signal (Figure 6).

A report by Zhang and Tanner (2021) using SARS-CoV-2 and the influenza virus highlighted the suitability of DARQ-LAMP for multiplexing [36]. This study included four targets: the E gene (SARS-CoV-2), influenza A, influenza B, and a human gene as an internal control (beta-actin (ACTB)), labeled with JOE, Cy5, FAM, and ROX fluorophores, respectively. Sensitivity was not affected by the inclusion of up to four primer sets in the reaction compared with singleplex reactions. However, standard curves with different viral RNA concentrations were not performed. The reactions were conducted with 50 copies of SARS-CoV-2, a 1:10,000 dilution for influenza A, and 21 copies for influenza B [36].

DARQ-LAMP multiplexing has also been applied to detect FMDV (foot-and-mouth disease virus), vesicular stomatitis virus (VSV), and bluetongue virus (BTV) in cattle [37]. In this approach, Fd was labeled at the 3′ end with FAM for FMDV, Cy5 for VSV, and Cy3 for BTV. Fan et al. achieved nearly 100% sensitivity and specificity for detecting these three viruses. However, the LoD values were 2477 copies for FMDV, 525 copies for VSV, and 913 copies for BTV, which are significantly higher than those obtained with RT-qPCR. These results might be due to the inclusion of three primer sets in the reaction, leading to potential primer interactions, which is a known limitation of multiplex assays [37].

DARQ-LAMP multiplexing has also been applied in avian virus detection, targeting three chicken viruses: chicken parvovirus (ChPV) with an LoD of 304 copies, chicken infectious anemia virus (CIAV) with an LoD of 749 copies, and fowl adenovirus serotype 4 (FAdV-4) with an LoD of 648 copies [38].

This approach has also been employed in singleplex reactions for detecting pathogens such as helminth worms (*Schistosoma mansoni* and *Strongyloides*) [79], as well as bacterial infections, including *Salmonella* [80], methicillin-resistant *Staphylococcus aureus* (MRSA) [81], and *Brucella* [82].

### 3.7. Quenching of Unincorporated Amplification Signal Reporters (QUASR)

Similar to DARQ, QUASR utilizes primers labeled with a fluorophore at the 5′ end, which can either be FIP/BIP or LPF/LPB, and includes a probe attached with a quencher molecule at the 3′ end. The probe has a short sequence (7–13 bases) that is complementary to the 5′ end of the labeled primer, with a melting temperature (Tm) about 10 °C below the LAMP reaction temperature. Additionally, the quencher probe is added in higher concentrations (1.5×) compared with the labeled primers. Under these conditions, the primers and probe remain dissociated during amplification. After the reaction (35–40 min), the tubes are cooled to room temperature. At this stage, primers that were not incorporated in the reaction hybridize with the quencher probe, preventing fluorescence (“dark”). In contrast, primers that bind to the target sequence are unavailable to bind with the quencher, allowing fluorescence to be detected (“bright”) as a result of successful amplification [40] (Figure 7).

Ball et al. demonstrated the use of this method on three RNA viruses: the West Nile virus (WNV), Chikungunya virus (CHIKV), and bacteriophage MS2 (used as a model for viral RNA). For MS2, the FIP was labeled with Cy5; for WNV, the FIP was labeled with ROX; and for CHIKV, the BIP was labeled with FAM. After amplification, the samples were visualized under UV light: WNV-positive samples appeared red, CHIKV-positive samples green, and samples positive for both viruses appeared yellow [40].

However, a limitation of the method is the overlapping of colors, which can lead to bias in result interpretation, and quantification remains a challenge. Prye et al. introduced a solution for this by developing a POC system that couples QUASR with a smartphone-controlled application that uses an algorithm for detecting luminescence and performing image analysis. The study demonstrated multiplex detection of the Zika virus (ZKV), CHIKV, and dengue virus (DENV). ZKV was labeled with Cy5, CHIKV with FAM, and DENV was detected using the intercalating dye SYTO9. The LoD for ZKV was 44 copies/mL, while for CHIKV, it ranged from 10^8^ to 10^3^ PFU/mL, with detection times of 7 to 15 min, respectively. DENV was detected in under 40 min [41].

### 3.8. Fluorescence of Loop Primer upon Self-Dequenching LAMP (FLOS-LAMP)

This methodology utilizes a self-dequenching system in which quenching occurs in the unbound state, with bases that can act as quenchers (Figure 8). Gadkar applied FLOS-LAMP for detecting the Varicella Zoster virus, a herpesvirus. The approach involved designing a probe where LPB was conjugated to the FAM fluorophore. In this strategy, a T base of each LAMP primer (F3, FIP, BIP, and LPB) was labeled with the FAM fluorophore, without the use of quenchers. Among these, LPB exhibited the best performance. The authors established criteria for selecting which T residue to label for quenching/dequenching effects, including the presence of cytosine (C) or guanine (G) at the 3′ terminal end and the position of the T base at the second or third position from the 3′ end. Once the primers hybridize to the target, the quenching effect is removed, and the fluorescence signal is detected. This method is not based on Förster Resonance Energy Transfer (FRET) but rather on the quenching effect caused by cytosine and guanine [39].

Similarly, Hardinge and Murray (2019) developed a variation using FIP and LPF labeled with JOE, TAMRA, or FAM, using fluorescent quenchers. The hypothesis is that during amplification, the formation of concatemer structures keeps the fragments labeled with fluorophores in proximity, allowing FRET to occur, leading to energy transfer and a quenching effect [83]. In their work, fluorescence analysis was combined with SYTO9, a dsDNA intercalant, to monitor amplification and detect false positives, specifically for identifying the mosaic virus. SYTO-9 and SYTO-82 have better performance than EvaGreen and SYBR Green in LAMP reactions [83].

### 3.9. Molecular Beacon Method

In the context of viral identification, viruses such as HIV, HBV, HCV, HEV, and others require urgent application in POC methods. Assays that enable the simultaneous detection of multiple pathogens while providing viral load information offer a significant advantage, such as those utilizing molecular beacons (Figure 9). Nyan and Swinson applied molecular beacons to identify HIV, HBV, HCV, HEV, DENV, and WNV in a multiplex reaction [42]. In this assay, the LPB were bi-labeled with a fluorophore at the 5′ end and a quencher at the 3′ end (probe). The fluorophores used included FAM, Texas Red, and TET, with BHQ1 or BHQ2 (black hole quencher 1 and 2) as the quenchers. Before amplification, the quenchers and fluorophores are in close proximity, suppressing fluorescence. As amplification proceeds and the probe hybridizes with the target, the distance between the fluorophore and quencher increases, causing FRET (Förster Resonance Energy Transfer) to decrease. This results in an increase in fluorescence, allowing for signal detection. Visualization was achieved using UV light excitation and by comparing the pattern of bands in agarose gel. This assay demonstrated 97% sensitivity and 100% specificity [42].

As is commonly known, LAMP can be prone to non-specific amplification, and the inclusion of multiple primers can lead to the formation of self-dimers and heterodimers, making multiplex reactions with more than 3 or 4 targets particularly challenging. Visual analysis with UV (naked eye), especially in multiplex reactions, must be conducted carefully to minimize the risk of false positives in diagnostics.

For SARS-CoV-2, molecular beacons were applied in an RT-LAMP multiplex assay to detect the ORF1ab and N genes, labeled with FAM and NED, respectively [43]. The LoD for this assay was 20 and 2 copies/μL. In the same study, One Step Displacement (OSD) probes were employed to identify a single-site mutation (wild type and P681R) in the S gene. The first OSD probe had a sequence complementary to the target and was labeled with a 5′ fluorophore (FAM or NED), while the second probe had a quencher at the 3′ end (BHQ1 or BHQ2) and hybridized with the opposite strand of the first probe. During amplification, the separation of the quencher and reporter occurred, leading to the release of fluorescence and signal capture. The OSD probes were capable of detecting 2 × 10^6^, 2 × 10^5^, and 2 × 10^4^ copies/μL of RNA for the wild-type, P681H, and P681R variants, respectively [43].

Reviewing fluorescence-based LAMP assays for viral detection reveals a significant gap in their application for the detection of viruses from the Herpesviridae family. This family is of critical importance in hematopoietic stem cell transplantation (HSCT), as infections can lead to severe complications and early identification is crucial. The complexity of Herpesviridae infections, particularly during reactivation in immunocompromised patients, underscores the need for rapid, sensitive, and specific diagnostic methods. Given the challenges in detecting these viruses, the following sections will explore how LAMP can be effectively applied to detect viruses within this family, potentially bridging this diagnostic gap.

## 4. LAMP in Herpesviridae Detection

As described previously, LAMP is an approach widely used to detect pathogens, such as bacteria and viruses. This applicability was noted in the SARS-CoV2 pandemic; a lot of papers have been published in order to demonstrate LAMP’s potential in viral detection [84,85].

Herpesviridae is a family of viruses known for their ability to establish lifelong infections in their hosts. These viruses are widespread in the animal kingdom, infecting a variety of species, including mammals, birds, reptiles, and fish. The family is divided into three subfamilies: Alphaherpesvirinae, Betaherpesvirinae, and Gammaherpesvirinae [86,87]. In general, the viral genome is composed of linear double-strand DNA, packaged into a capsid. Lytic transcription is classified in three phases: immediately early (IE), early (E), and late (L). IE genes products are responsible for transcription activation, E genes codify DNA replication proteins and L genes produce viral capsid [10].

Among these, three viruses can be highlighted with great clinical relevance in infections in humans as Cytomegalovirus, Epstein–Barr, and HHV-6. In fact, these viruses have great clinical relevance in the infection of patients undergoing organ and hematopoietic stem cell transplants [88,89]. Therefore, it is of great importance to develop methodologies that allow the rapid and effective identification of these viruses in this context.

However, specific considerations should be made for herpesviruses, particularly regarding their lytic and latent phases and the availability of viral particles in different sample types. During the latent phase of infections such as the Cytomegalovirus (CMV), Epstein–Barr Virus (EBV), and Human Herpesvirus 6 (HHV-6), viral DNA resides mainly within infected cells. In this phase, whole blood is more sensitive for detecting viral loads, as it contains intracellular viral DNA. Thus, whole blood is the preferred sample type for monitoring latent infections, where viral particles are not actively released into the bloodstream. Conversely, during the lytic phase, when the virus reactivates and begins replication, it releases a higher number of viral particles into the bloodstream. In this phase, plasma is expected to contain more viral particles, reflecting the extracellular virus. Therefore, plasma may be a better indicator of active viral replication during reactivation. This differentiation between sample types also impacts the limit of detection (LoD) and sensitivity of assays, with whole blood providing a more sensitive measure during latency and plasma being more effective for detecting active infection due to higher extracellular viral loads. Viral load dynamics also play a critical role in detection. During early replication, viral loads are often lower, potentially leading to false-negative results if samples are collected too early [90,91].

Thus, the timing of collection is critical for achieving high sensitivity in viral detection. Kraft et al. (2012) highlighted the substantial variability in viral load quantification across different laboratories, with results ranging from 100,000 copies/mL to 100 copies/mL for the same patient using gold standard method qPCR [92]. This highlights the need for standardization in viral load measurement to reduce discrepancies and improve clinical decision-making. These factors—variability in sample type, viral load dynamics, and inter-laboratory differences—pose challenges for viral load quantification in assays like LAMP and qPCR. Standardization efforts are crucial to improve the accuracy of these metrics and ensure more reliable detection across different clinical settings [91,92].

In Table 2, a summary of the publications on the application of LAMP for identifying Herpesviridae viruses is shown.

### 4.1. Cytomegalovirus

Cytomegalovirus (CMV) is a member of the Herpesviridae family, specifically belonging to the Betaherpesvirinae subfamily. This virus is globally distributed and infects a wide range of vertebrates, with humans being the primary host. While CMV infections are often asymptomatic in healthy individuals, they can cause severe complications in immunocompromised individuals and during pregnancy [110,111,112]. In hematopoietic stem cell transplantation (HSCT) patients, CMV reactivation can lead to severe conditions like pneumonia, which has a mortality rate of over 50% and requires urgent care [15].

The CMV genome is the largest among all human herpesviruses, consisting of double-stranded DNA measuring 230 to 240 kilobases. This genome is enclosed in an icosahedral capsid, which is surrounded by an envelope derived from the host–cell. Human CMV is primarily transmitted through close contact with bodily fluids, such as saliva, urine, blood, and breast milk, and can also be transmitted through organ transplantation and blood transfusions [113].

For CMV detection, most commercial kits are based on RT-qPCR technology; however, the literature also supports LAMP as a viable methodology [95]. A recent study [96] used qLAMP for large-scale screening of newborns for congenital CMV infection, with a dynamic range of 1.1 × 10^8^ to 1.1 × 10^3^ copies/μL. The authors demonstrated that a colorimetric LAMP assay could detect CMV at levels as low as 1.1 × 10^3^ copies/μL within just 30 min.

In another study, Reddy et al. [93] analyzed 40 patients with viral retinitis and successfully detected CMV using LAMP, with a sensitivity of 10 copies/μL of CMV DNA. A cross-comparison with quantitative PCR showed 100% agreement between the two techniques, highlighting LAMP’s applicability. Similarly, the sensitivity and specificity of the LAMP technique were evaluated in a study involving 20 post-hematopoietic stem cell transplantation children. The sensitivity, specificity, positive predictive value, and negative predictive value of CMV LAMP for detecting 500 copies/tube (based on qPCR quantification) were 80.0%, 98.9%, 66.7%, and 99.4%, respectively [94].

### 4.2. Epstein Barr Virus

The Epstein–Barr virus (EBV), also known as Human Herpesvirus 4 (HHV-4), is a member of the herpesvirus family. It was discovered in 1964 by Michael Anthony Epstein and Yvonne Barr and is one of the most common human viruses, widely distributed globally. EBV is primarily transmitted through the exchange of bodily fluids, especially saliva, though it can also be transmitted through blood and genital secretions. Immunocompromised individuals, such as those with HIV/AIDS or those undergoing organ transplantation, are at a higher risk of developing complications associated with EBV infection [114,115].

For the detection of EBV, similar to CMV, quantitative PCR techniques are the primary diagnostic methods used in clinical settings. However, there are few studies exploring the efficacy of detecting this virus using LAMP. One of the earliest studies was published in 2006 by Iwata et al. [97], which demonstrated that diagnosing initial EBV infection using LAMP exhibited a sensitivity of 86.4% and a specificity of 100%. In comparison, a real-time PCR assay showed a sensitivity of 84.1% and a specificity of 98.4%. Over the course of the study, a longitudinal analysis revealed that the detection rate of EBV DNA in serum declined through the LAMP assay, correlating with a reduction in EBV load. Notably, no EBV DNA was detectable in the serum 40 days after the onset of symptoms.

Similarly, LAMP-based detection of EBV was evaluated for the early diagnosis of nasopharyngeal carcinoma, a cancer strongly associated with EBV infection. Using 33 samples, LAMP detection demonstrated high specificity. Compared with PCR, LAMP is simpler, more convenient to perform, and does not require specialized equipment, making it more cost-effective and practical [116].

### 4.3. Herpes Simplex Virus

Herpes Simplex Virus (HSV) is a common and highly contagious virus belonging to the herpesvirus family. There are two main types: HSV-1 and HSV-2, each with tropism for specific anatomical regions. HSV-1 is primarily associated with oral infections, while HSV-2 tends to cause genital infections. However, both can infect either area and establish latency in the sensory ganglia following the initial infection. HSV is primarily transmitted through direct contact with the mucous membranes or broken skin of an infected person, including through activities such as kissing, oral–genital contact, and sexual intercourse. In immunocompromised individuals, HSV reactivation can lead to severe complications such as hepatitis, encephalitis, or meningitis. Diagnosis of HSV is typically based on clinical presentation, but viral culture, PCR, and serological tests can confirm the infection [117,118].

In the literature, there is a broader body of research on the applicability of LAMP for detecting HSV using different biological samples [98,99,100]. However, for the purposes of this discussion, we will focus on HHV-6 (A and B), HHV-7, and HHV-8, which are of particular significance in the context of organ transplantation and hematopoietic stem cell transplantation (HSCT) [119].

### 4.4. Human Herpesvirus 6 and Human Herpesvirus 7

Human Herpesvirus 6 A (HHV-6A), Human Herpesvirus 6 B (HHV-6B), and Human Herpesvirus 7 (HHV-7) are Betaherpesviruses (genus Roseolovirus). HVV-6A and HHV-6B showed 90% identity, and genomic variations are mainly in IE1 (immediate early region 1), glycoproteins B (gB), and H (gH) genes. Infection of these viruses are acquired in childhood; their manifestation causes exanthem subitum, which is characterized by skin rashes, fever, and gastrointestinal and respiratory infection. HHV-6 infections are more common than HHV-7. HHV-6 typically manifests around 6 to 24 months of age, while HHV-7 manifests between two and five years old [120].

During infection, HHV-6 replicates in several cell tissues such as the central nervous system (CNS), salivary glands, and lymph nodes, while HHV-7 replicates in lymphoid tissues, tonsils, and skin tissues. A latency mechanism is established in HHV-6 in monocytes and macrophages cells and T lymphocytes in HHV-7 [121,122].

In HSCT, there is a high risk of reactivation of HHV-6 in the first four weeks after transplant. Complications of HSCT include fever, rash, and hematopoietic alterations such as decreases in granulocyte and macrophage cells. The most critical complication is encephalitis, which affects around 11.6% of patients [18]. However, this condition should be monitored and viral load evaluated, because of high morbidity in these patients.

For HHV-6B, Yoshikawa et al. [101] demonstrated that LAMP was highly sensitive (94.0%) and specific (96.0%) for its detection using serum samples obtained from febrile children. The assay was performed with dry reagents maintained at 4 °C to avoid refreezing steps and reagents degradation. In another work, considering the monitoring of active HHV-6 infection in hematopoietic stem cell transplant recipients, LAMP also showed high levels of sensibility and specificity positive predictive value and negative predictive value, with results of 80%, 100%, 100%, and 97.8%, respectively. The assay had LoD between 10 and 100 copies per tube and showed a high correlation between viral load and turbidity, with R^2^ = 0.969 and R^2^ = 0.979 [123]. However, the correlation coefficient between LAMP and viral load as measured by qPCR was lower, at R^2^ = 0.342. The correlation improved slightly with additional DNA extraction, yielding an R^2^ = 0.4019, but it was still insufficient to establish a strong correlation between LAMP and qPCR in this study [123].

LAMP was also used to discriminate between HHV-6A and HHV-6B. Using the serum of 20 patients, the assay proved to be highly effective in differentiating the variants, and also proved to be a faster and cheaper tool to be used clinically. This is important considering the diagnosis of neurological diseases [104]. Ihira et al. investigated HHV-6 infection in children aged 5 to 18 months. LAMP was able to detect the virus at 50 copies/tube, and with the authors increasing the primer concentration, the LoD improved two-fold, to 25 copies/tube [102]. In another evaluation of LAMP efficiency to detect HHV-6 using patient’s serum, the sensitivity, specificity, positive predictive value, and negative predictive value of the HHV-6 LAMP method without DNA extraction were 95.5%, 95.2%, 94.0%, and 96.4%, respectively [103].

HHV-7 detection was also accessed using turbidity for the detection of LAMP products and resulted in a sensitivity of 500 and 250 copies/tube for 30 and 60 min reactions, respectively. In the same article, the authors also concluded that, since a turbidity assay is less sensitive than agarose gel electrophoresis, no HHV-7 LAMP product could be detected in plasma samples after a 30 min LAMP reaction. After a 60 min LAMP reaction, HHV-7 LAMP product could be detected in acute-phase plasma samples [105].

### 4.5. Human Herpesvirus 8

Human Herpesvirus 8 (HHV-8), also known as Kaposi’s sarcoma-associated herpesvirus (KSHV), was first isolated in 1994 from biopsies of Kaposi’s sarcoma (KS) in an AIDS patient. KS is a lymphoproliferative disease that primarily affects the lymphatic system and blood vessels, with the first manifestations often appearing as skin neoplasms. HHV-8 infection drives the expression of interleukin 6 (IL-6), leading to symptoms such as splenomegaly, and is associated with other diseases like primary effusion lymphoma (PEL) and Multicentric Castleman’s disease (MCD) [124]. Transmission occurs through saliva, sexual contact, and organ transplantation [125]. HHV-8 infections predominantly affect immunocompromised individuals, particularly those with HIV/AIDS. AIDS-associated KS (AIDS-KS) is significantly more common than other forms of KS, such as iatrogenic or classic KS. Interestingly, HHV-8 infection is most persistent in Central, East, and South Africa (with prevalence rates exceeding 50%) and is also prevalent in Mediterranean countries and the Middle East (5–20%) [126,127].

In the latent phase, HHV-8 primarily infects B cells, neoplastic KS spindle cells, and atypical endothelial cells in KS lesions. The lytic phase is more common in MCD, where HHV-8 infects B cells, and IL-6 drives their proliferation.

Reactivation of HHV-8 is more commonly reported in solid organ transplantation (SOT) than in hematopoietic stem cell transplantation (HSCT). KS is approximately 200 times more frequent in SOT recipients compared with the general population. In HSCT, KS has been reported in 32 cases between 2004 and 2017, with a prevalence of 0.05% in autologous HSCT and 0.17% in allogeneic HSCT [128]. As a result, pre-transplant screening for HHV-8 is common in endemic regions [128]. Detection of HHV-8 using LAMP has demonstrated high sensitivity, with a detection limit of 100 copies of the target sequence per tube [106].

### 4.6. Varicella Zoster

The Varicella Zoster Virus (VZV) is a member of the Alphaherpesvirus subfamily. It exclusively infects humans and is primarily transmitted through airborne saliva via coughing, sneezing, or through direct contact with skin lesions. VZV primarily targets T lymphocytes, epithelial cells, and ganglionic neurons. The first infection causes varicella (chickenpox), after which the virus enters a latency phase in ganglionic neurons. Virus recurrence, known as shingles, typically occurs in immunocompromised individuals and older adults due to declining antibody levels. In addition to shingles, severe complications such as meningoencephalitis, hepatitis, and pancreatitis can also occur [129].

In HSCT patients, VZV reactivation can lead to complications such as postherpetic neuralgia in approximately 25% of cases, with a mortality rate of 9.7% [130]. Some individuals may also develop mild infections, including skin rashes, caused by the VZV Oka vaccine strain. In this context, LAMP assays have proven effective in distinguishing between wild-type VZV infection and reactivation of the latent VZV Oka vaccine strain, aiding in treatment decisions [108].

Okamoto et al. and Higashimoto et al. applied LAMP to differentiate between the two VZV strains by detecting mutations in ORF 62. Both groups successfully amplified targets from 50,000 to 500 copies and 50,000 to 100 copies, respectively [107,108]. More recently, Higashimoto et al. published comparative results of LAMP detection for VZV and vaccine reactivation, achieving sensitivities of 93.3% and 84.4%, respectively [109]. Methodologies utilizing fluorogenic probes, as discussed earlier, can also be applied for VZV detection, showing strong potential to identify herpesviruses through fluorescence-based assays [39].

## 5. Discussion

Considering the data discussed for the use of LAMP in viral detection, especially in Herpesviridae, it is possible to emphasize the effectiveness of the methodology in detecting a wide range of viruses. There is a gap in Herpesvirus detection using the LAMP methodology, with few published studies about these targets. Currently, several papers have been published using LAMP for different pathogens, highlighting viral detection, specifically for SARS-CoV-2. However, it should be noted here that this methodology proved to be faster and cheaper compared with the standard methodology currently used. Among other advantages of the LAMP, one can mention the broad ranges of pH and temperature acceptable for the reaction and the versatility in reading methods, which simultaneously maintain the sensitivity and specificity of the experiment similar to the PCR methodology [131].

On the other hand, the results of LAMP-mediated amplification can be confirmed by different methods, such as changes in turbidity and fluorescence alterations, which can affect the sensitivity and versatility of the method. Confirmation through colorimetric assay, for instance, involves using a DNA intercalating dye, thereby increasing the risk of contamination of LAMP reaction solutions, given that the assay requires tube opening after the reaction has been completed for the addition of the intercalating solution [132]. Additionally, it is worth noting that this assay exhibits some degree of subjectivity, considering that the analysis of the result is based on observing the color change in the tubes under ambient light. This factor varies depending on the observer and the environment in question [133].

A study conducted by Aoki et al. in 2021 demonstrated that in SARS-CoV-2 analyses, the sensitivity of diagnosis by colorimetric assay depends on the observer’s interpretation of colors and the viral load of the sample [133]. The study used 62 clinically confirmed SARS-CoV-2 samples, verified by RT-qPCR. When tested using RT-LAMP colorimetric assay, it was observed that 79% of these samples yielded positive results, 14.5% were categorized as indeterminate, and 6.5% were false-negative. It is worth noting that all positive and indeterminate samples, regardless of RT-LAMP colorimetry, exhibited amplification products when visualized on agarose gel, demonstrating that the color change may not occur despite amplification. This indicates that, although colorimetric assays confirm LAMP-mediated amplification and this method is sensitive for identifying various pathogens, such as SARS-CoV-2, its sensitivity is lower when compared with alternative methods, representing a methodological vulnerability in diagnosing the disease [133].

While traditional LAMP or colorimetric methods are excellent options for point-of-care (POC) diagnostics when detecting the mere presence or absence of a pathogen, they are insufficient for herpesvirus detection, where precise viral quantification is essential. Colorimetric and turbidity-based methods lack the necessary precision. Instead, fluorescence-based LAMP provides a viable alternative, offering greater accuracy in quantification. For instance, Yaren et al. demonstrated that RT-LAMP could achieve a LoD of 10 copies/μL for SARS-CoV-2, compared with 25 copies/μL for RT-qPCR [31]. Nyan et al. [42] successfully multiplexed six viral pathogens using the molecular beacon method, correlating RFU with UV intensity across serial dilutions. Multiplexing multiple targets in an assay enhances both the reliability and cost-effectiveness of diagnostics.

However, LAMP multiplexing presents challenges, such as the interaction of multiple oligonucleotides, which complicates optimization. Strategies to mitigate the formation of secondary structures, such as dimers (self-dimers, heterodimers, and hairpins), include adding betaine (0.1 to 1 M). Magnesium concentration is also critical for successful amplification, as it acts as a cofactor for enzyme function and may require optimization [2,134]. Additionally, sensitivity may decrease compared with singleplex reactions due to competition for amplification resources. Careful design of probes and modified primers is essential to avoid mismatches and impaired reactions. The selection of fluorophores should also be made with caution to prevent overlapping wavelengths.

In the context of hematopoietic stem cell transplantation (HSCT), managing CMV, EBV, and HHV-6 is critical due to the severe complications these viruses can cause post-transplant, including graft loss or death. Accurate viral quantification is, therefore, essential for guiding appropriate antiviral treatment. However, fluorescence-based LAMP methods often require sophisticated detection systems, which can limit their broader application. Viral detection depends on several factors such as the virus phase and its abundance in whole blood or plasma. Furthermore, assay design is fundamental for accurate detection. However, advances in digital droplet technologies integrated with microfluidics [5] and lateral-flow biosensors incorporating CRISPR/Cas technology offer potential improvements for detecting LAMP reactions [6]. These innovations could overcome current limitations and enable point-of-care (POC) applications [135].

Despite these challenges, LAMP variations, particularly when combined with lateral-flow assays and advanced fluorescence techniques, remain promising options for enhancing the sensitivity and specificity of viral detection methods. Additionally, the emergence of new, more accessible instruments for POC diagnosis monitoring by smartphones and the use of recombinant enzymes can serve as alternatives to overcome these limitations, making herpesvirus diagnosis more accessible [70]. Consequently, LAMP is a viable choice for POC diagnostics, especially for the rapid detection of Herpesviridae viruses, contributing to more reliable diagnoses.

## 6. Conclusions

The LAMP technique offers a variety of adaptations tailored for numerous applications, including mutation identification and pathogen detection. However, in the diagnosis of herpesviruses, traditional LAMP approaches remain relatively underexplored. Methods involving multiple targets or the integration of fluorogenic probes are also areas ripe for further investigation. These advanced variations hold significant promise, particularly because they could enable the quantification of viral load, an essential factor in clinical diagnostics. Nonetheless, certain challenges persist. These include optimizing reaction conditions, improving the limit of detection, addressing amplification inhibition caused by primers and probes, and refining labeling molecules to enhance specificity in LAMP assays. Despite these hurdles, ongoing advancements in biotechnology, particularly with the development of novel enzymes and reagents, are paving the way for LAMP to become an even more robust and reliable technique. As it evolves, LAMP could potentially offer faster, more cost-effective, and equally reliable alternatives to traditional methods. While qPCR continues to be the gold standard for quantification due to its precision and widespread validation, the landscape could shift with new innovations, making LAMP a powerful tool for the detection and management of Herpesviridae, and potentially other pathogens, in the future. These advancements could also improve clinical monitoring of viral load, making it easier and more precise for healthcare professionals to manage patient care, ultimately benefiting patients by providing more accurate viral load measurements and treatment decisions.

## 7. Glossary

Assimilation Probe Technique or Displacement Probe Method: In this method, the loop primers (either loop primer forward or loop primer backward) are modified with a fluorophore at one end and a universal sequence (sequence common for all probes) at the other. A complementary oligonucleotide probe (called the quencher strand) with a quencher molecule is included. Initially, the quencher binds to the universal sequence and quenches the fluorescence. As the amplification proceeds, the newly synthesized DNA displaces the quencher strand, releasing the fluorophore and resulting in fluorescence, which indicates the amplification of the target sequence.

DARQ-LAMP (Detection of Amplification by Releasing of Quenching) is a variant that utilizes a fluorophore attached to the 5′ end of an inner primer (usually FIP or BIP) and a complementary quencher probe. During amplification, the quencher is displaced, and fluorescence is released, allowing for real-time monitoring of the reaction.

FLOS-LAMP (Fluorescent Loop-mediated Isothermal Amplification of Self-quenching) involves self-quenching fluorophores. In this technique, fluorophores are attached to specific nucleotide bases of LAMP primers. When the primers bind to the target, the quenching effect is relieved, and fluorescence is emitted, allowing for easy detection. This method does not require external quenching probes, simplifying the assay design.

Molecular Beacons: are hairpin-shaped oligonucleotide probes with a fluorophore and a quencher at opposite ends. In the absence of the target, the beacon is in closed conformation, bringing the quencher and fluorophore in proximity to prevent fluorescence. When the beacon hybridizes with the target, it undergoes a conformational change, separating the quencher from the fluorophore, and allowing fluorescence emission.

QUASR-LAMP: QUASR (Quenching of Unincorporated Amplification Signal Reporters) is a method that uses a quencher probe complementary to a short region of the LAMP primer. The probe remains in solution unbound during the reaction. After amplification, the reaction is cooled, and unincorporated primers hybridize with the quencher, leaving only the amplified target detectable by fluorescence.

## Figures and Tables

**Figure 1 ijms-25-11506-f001:**
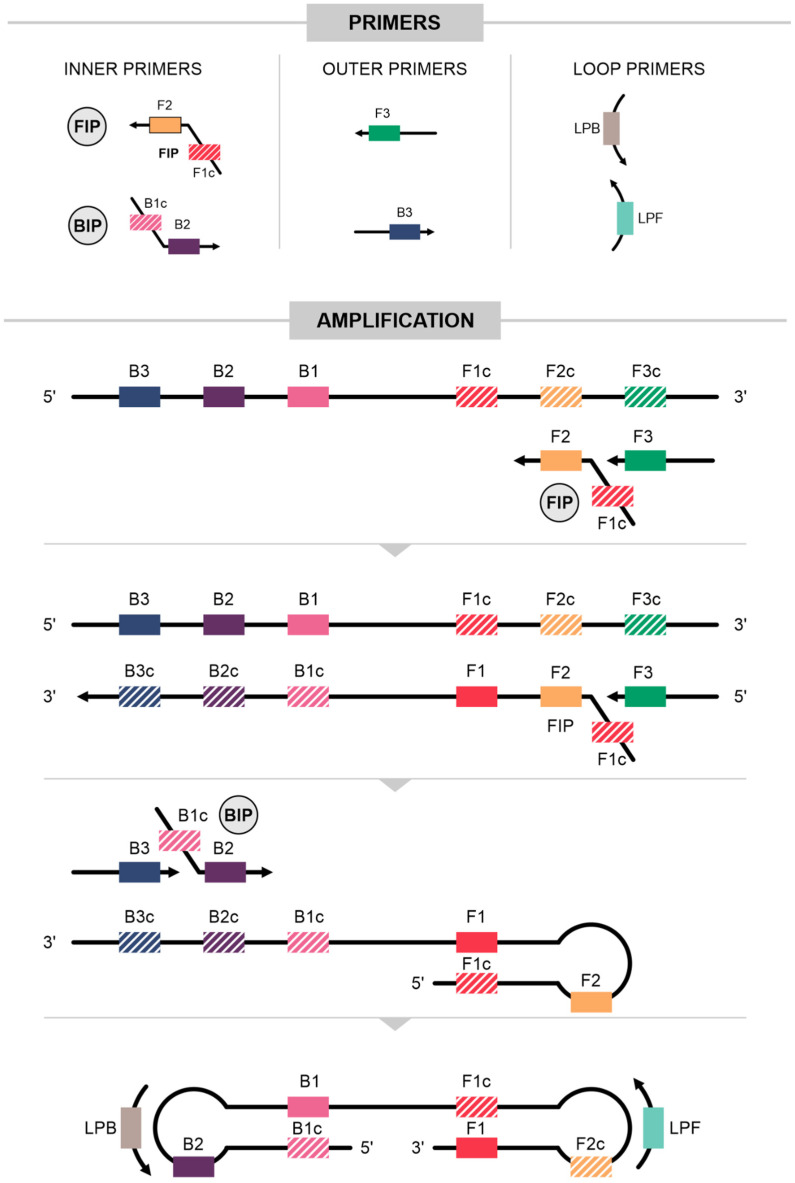
Diagram of loop-mediated isothermal amplification (LAMP) reaction. In a traditional LAMP assay, 4 to 6 primers are included, comprising outer primers (F3 and B3) and loop primers (LPF and LPB). Initially, FIP and F3 attach to the target sequence and initiate DNA strand synthesis, facilitated by Bst polymerase. Following this, BIP and B3 bind to their complementary targets and synthesize DNA on the opposite strands. The amplicons, with their complementary regions, undergo self-hybridization, leading to the formation of characteristic dumbbell-shaped structures.

**Figure 2 ijms-25-11506-f002:**
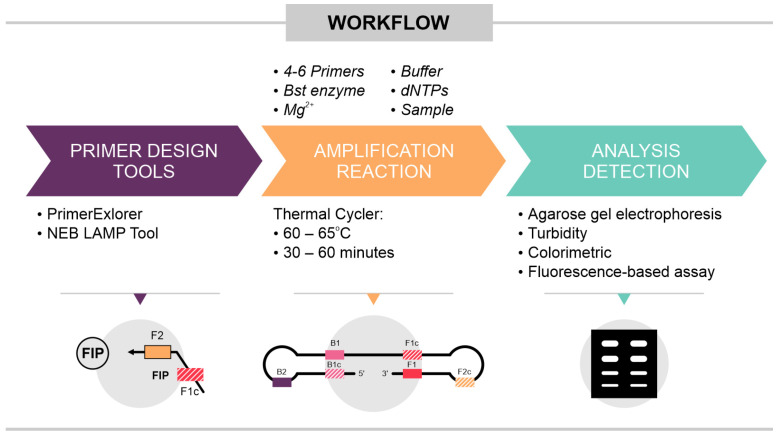
Illustrates the workflow of the LAMP reaction, highlighting three key points. Primer Design: primers must be carefully designed to ensure specificity for the target sequence. Optimization of the LAMP Reaction: the reaction requires optimization of several parameters, including primer concentrations, the temperature for Bst (*Bacillus stearothermophilus*) polymerase activity, and the amplification time. Detection Methods: detection of LAMP products can be carried out in various ways, such as visualization on an agarose gel, detection of turbidity due to the release of pyrophosphate, colorimetric detection via color change (e.g., phenol red), or fluorescence-based detection using labeled probe.

**Figure 3 ijms-25-11506-f003:**
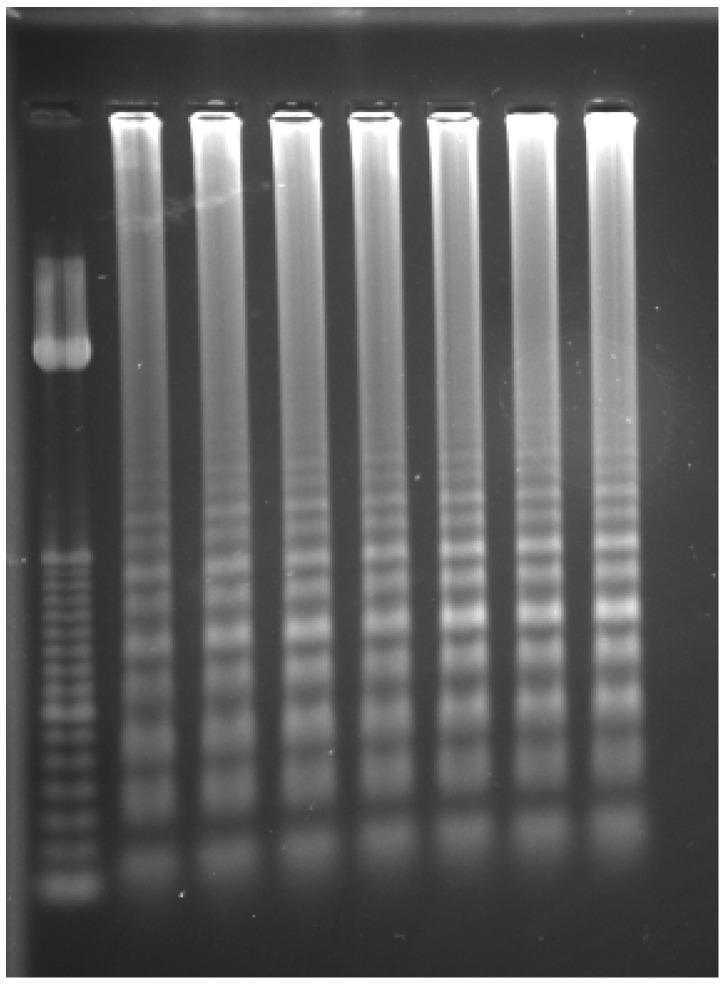
Electrophoresis detection of LAMP. Illustration showing the differences between a molecular weight ladder and LAMP amplicon patterns. Left: molecular weight-size marker (ladder) used to estimate amplicon sizes. Right: gel showing seven LAMP amplicon samples.

**Figure 4 ijms-25-11506-f004:**
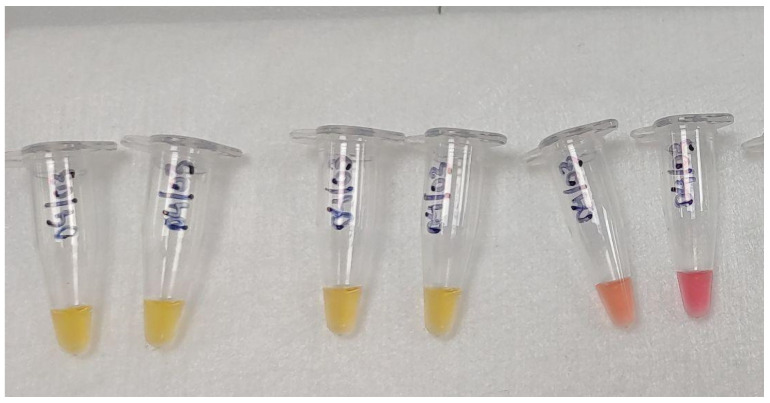
Colorimetric method. An example of the naked-eye visualization of color changes with a phenol red pH indicator. During the amplification process, protons are released into the solution, resulting in a pH decrease. Thus, in positive reactions, the color of the reaction solution turns yellow (left), while in negative reactions it remains pink (right).

**Figure 5 ijms-25-11506-f005:**
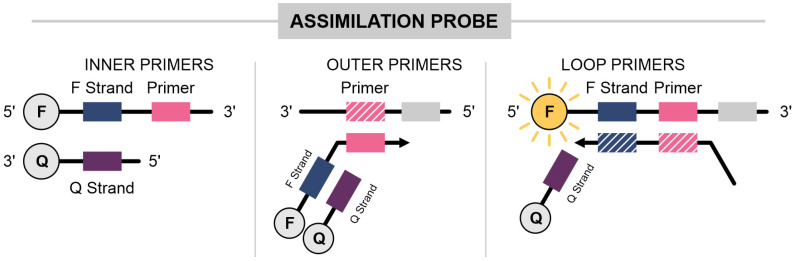
Assimilation probe (displacement probe). Primers are customized with a universal sequence (F strand) and a fluorophore attached at the 5′ end. An oligonucleotide containing a quencher is complementary to the universal sequence (Q strand). Prior to the start of the reaction, the probe and the Q strand remain bound. As the amplification begins and Bst strand displacement occurs, the quencher strand is released, resulting in signal amplification. The gray blocks represent the surrounding regions of the oligonucleotide target, while the pink and blue dotted blocks indicate regions complementary to the oligonucleotide target.

**Figure 6 ijms-25-11506-f006:**
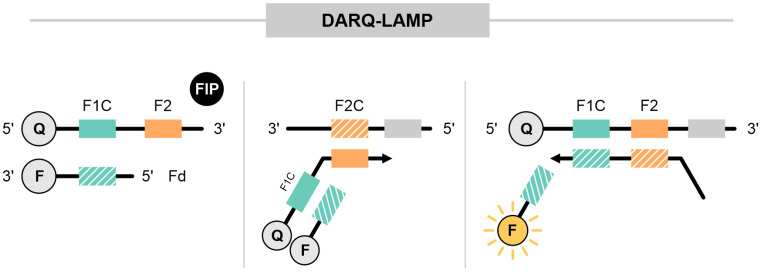
Detection of amplification by releasing of quenching (DARQ-LAMP). The FIP was modified with the attachment of a quencher molecule at the 5′ end, while a probe complementary to the F1c region included a fluorophore at the 3′ end. The modified FIP and probe remained bound. During Bst activity, a new strand was synthesized, leading to the release of the probe and detection of the signal. The gray blocks represent the surrounding regions of the oligonucleotide target, while the green and orange dotted blocks indicate regions complementary to the oligonucleotide target.

**Figure 7 ijms-25-11506-f007:**
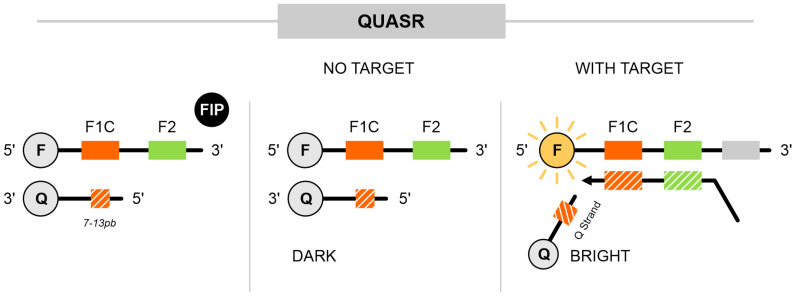
Quenching of unincorporated amplification signal reporters (QUASR). This approach includes a fluorophore at the 5′ end of either the FIP or BIP and incorporates a probe with a quencher at the 3′ end. The quencher probe is complementary to F1c only within a range of 7 to 13 base pairs, has a melting temperature approximately 10 °C lower than the LAMP reaction temperature (65 °C), and is included in the reaction at a higher concentration than the labeled primers. Thus, during amplification, primers and quencher probes remain in an unbound state. After the reaction, when the tubes are cooled, free primers and probes are in close proximity, resulting in no signal. However, when amplification occurs, primers are bound to the target, allowing the release of the signal and its detection. The gray blocks represent the surrounding regions of the oligonucleotide target, while the green and orange dotted blocks indicate regions complementary to the oligonucleotide target.

**Figure 8 ijms-25-11506-f008:**
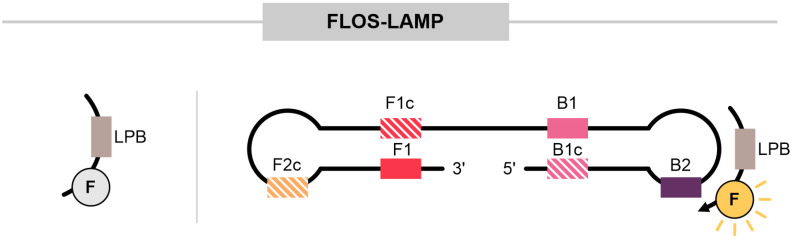
Fluorescence of loop primer upon a self-dequenching LAMP (FLOS-LAMP). In FLOS-LAMP, a fluorophore is attached to a T residue of one of the primers, without the need for a quencher. For adequate self-dequenching, the T residue chosen for fluorophore attachment must be adjacent to C (cytosine) or G (guanine) bases and other criteria. Here, we demonstrate the attachment in the LPB primer. When unbound, a self-quenching system inhibits signal release. During amplification, the labeled primer is incorporated into the reaction, leading to dequenching and the release of fluorescence.

**Figure 9 ijms-25-11506-f009:**
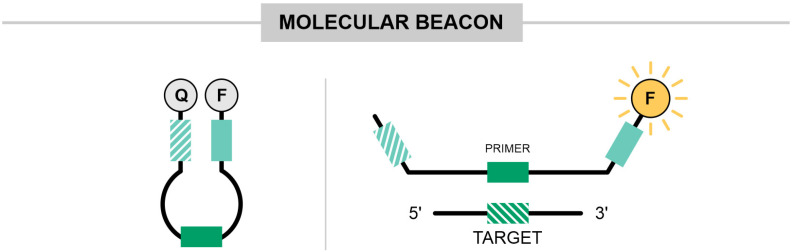
Molecular beacons. These primers are labeled with a fluorophore at the 5′ end and a quencher molecule at the 3′ end, forming a hairpin structure. In the absence of the target, the fluorophore and quencher remain in close proximity, preventing signal emission. During the amplification process, the target binding separates the quencher and fluorophore, resulting in fluorescence detection. The light green blocks represent regions complementary within the oligonucleotide, while the solid dark green block indicates the region of the oligonucleotide that is complementary to the target region.

**Table 1 ijms-25-11506-t001:** Summary of LAMP assay performed to detect various viruses.

Virus	Method	Target	Sample	LoD	Sensitivity	Specificity	Reference
ZKV	RT-LAMP	NS5	Urine	0.71 PFU (30 min)	-	-	Yaren et al., 2017 [30]
DENV	RT-LAMP	-	Urine	1.22 PFU (30 min)	-	-	Yaren et al., 2017 [30]
CHIKV	RT-LAMP	Nonstructural polyprotein gene	Urine	37.8 copies/reaction (30 min)	-	-	Yaren et al., 2017 [30]
SARS-CoV-2	DP-RT-LAMP	N and S	Heat-inactivated SARS-CoV-2	25 copies (12 min); 10 copies (16 min); 44 copies (16 min)	-	-	Yaren et al., 2021 [31]
SARS-CoV-2	DP-RT-LAMP	N and S	Saliva	200 copies	100%	-	Yaren et al., 2021 [31]
SARS-CoV-2	DP-RT-LAMP	N and S	Nasal swab	1000 copies 100 copies	100% 50%	-	Yaren et al., 2021 [31]
SARS-CoV-2	mRT-LAMP	RdRP and N	Nasopharyngeal swab	10^−5^/10^−6^ dilution (10-fold serial dilution)	96.92%	100%	Jang et al., 2021 [31]
SARS-CoV-2	mRT-LAMP	N gene (three regions)	Nasal swab	30 copies	87%	100%	Kline et al., 2022 [32]
FMDV	RRT-LAMP	3D	FMDV RNA template	100 copies/μL	-	-	Lim et al., 2020 [33]
PCV3	rLAMP	Cap	Plasmid DNA	50 copies/reaction	-	-	Kim et al., 2020 [34]
PEDV	LAMP-LFD	Spike	PEDV RNA	10 copies/μL	100%	100%	Areekit et al., 2022 [35]
SARS-CoV-2 and Influenza	DARQ-LAMP	E gene (SARS-CoV-2); IAV and IVB (Influenza)	Synthetic RNA	SARS-CoV-2 (50 copies); IAV (1:10,000 dilution); IAB (21 copies)	85%	100%	Zhang and Tanner, 2021 [36]
FMDV	mLAMP	3D	Vesicular fluid; Esophageal–pharyngeal; Vesicular skin	2477 copies/reaction (RNA standard)	95.5%	100%	Fan et al., 2022 [37]
VSV	mLAMP	Nucleoprotein N	RNA standard	525 copies/reaction	-	-	Fan et al., 2022 [37]
BTV	mLAMP	VP7	Whole blood	913 copies/reaction (RNA standard)	100%	100%	Fan et al., 2022 [37]
ChPV	mLAMP	NS	Cloacal swab; Heart; Kidney; Liver	304 copies/μL (plasmid)	100%	100%	Fan et al., 2023 [38]
CIAV	mLAMP	VP1	Cloacal swab; Heart; Kidney; Liver	749 copies/μL (plasmid)	100%	100%	Fan et al., 2023 [38]
FAdV-4	mLAMP	Hexon	Cloacal swab; Heart; Kidney; Liver	648 copies/μL (plasmid)	100%	100%	Fan et al., 2023 [38]
VZV	FLOS-LAMP	ORF62	Plasmid	500 copies/reaction	96.8%	100%	Gadkar et al., 2018 [39]
WNV	QUASR	E gene	Whole blood	10 PFU/reaction		-	Ball et al., 2016 [40]
CHIKV	QUASR	E1	Whole blood	-		-	Ball et al., 2016 [40]
DENV	QUASR	DENV 1-4	Viral strain	10^3.4^ copies/μL	-	-	Priye et al., 2017 [41]
ZIKV	QUASR	NS5-8640	Viral strain	44 copies/mL	-	-	Priye et al., 2017 [41]
CHIKV	QUASR	Env-984	Viral strain	10^8^–10^3^ PFU/mL	-	-	Priye et al., 2017 [41]
HIV	Molecular beacons	GAG	Plasma	10^2^ IU/reaction	97%	100%	Nyan and Swinson, 2015 [42]
HBV	Molecular beacons	P and S	Plasma	50 IU/reaction	97%	100%	Nyan and Swinson, 2015 [42]
HCV	Molecular beacons	5′NCR	Plasma	10^2^ IU/reaction	97%	100%	Nyan and Swinson, 2015 [42]
DENV	Molecular beacons	3′NCR	Plasma	10^2^ IU/reaction	97%	100%	Nyan and Swinson, 2015 [42]
WNV	Molecular beacons	5′NCR	Plasma	10^2^ IU/reaction	97%	100%	Nyan and Swinson, 2015 [42]
SARS-CoV-2	Molecular beacons	ORF1ab and N	Nasal and throat swabs	20 copies/μL (ORF1ab) 2 copies/μL (N gene)	-	-	Talap et al., 2022 [43]

Legend: LAMP: loop isothermal amplification mediated by loops; ZKV: Zika virus; RT-LAMP: reverse transcription LAMP; DENV: dengue virus; PFU: plaque forming unit; CHIKV: chikungunya virus; DP: displaceable probe-RT-LAMP; mRT-LAMP: multiplex RT-LAMP; FMDV: foot-and-mouth disease virus; RRT-LAMP: real-time reverse transcription LAMP; PCV3: porcine circovirus 3; rLAMP: real-time LAMP; PEDV: porcine epidemic diarrhea virus; DARQ-LAMP: detection of amplification by releasing of quenching LAMP; IAV: influenza A virus; IAB: influenza B virus; mLAMP: multiplex LAMP; VSV: vesicular stomatitis virus; BTV: bluetongue virus; ChPV: Chandipura virus; CIAV: chicken infectious anemia virus; FAdV-4: fowl adenovirus serotype 4; VZV: Varicella Zoster virus; FLOS-LAMP: fluorescence of loop primer upon self-dequenching LAMP; WNV: West Nile virus; QUASR: quenching of unincorporated amplification signal reporters; HIV: human immunodeficiency virus; HBV: hepatitis B virus; HCV: hepatitis C virus; IU: international units.

**Table 2 ijms-25-11506-t002:** Summary of LAMP applications in herpesvirus detection.

Virus	Method	Target	Sample	LoD	Sensitivity	Specificity	Reference
CMV	LAMP	gB gene	Vitreous from retinitis patients	10 copies/µL	95%	99%	Reddy et al., 2010 [93]
CMV	LAMP	gB gene	Whole blood	500 copies/tube	80%	98.9%	Suzuki et al., 2006 [94]
CMV	Commercial LAMP	Alethia CMV-LAMP^®^	Saliva	-	100%	99.8%	Izquierdo et al., 2023 [95]
CMV	LAMP (colorimetric); qLAMP	*U75*	Saliva and urine from newborns	1.1 × 10^3^ copies/µL (qLAMP 1.1 × 10^2^ (colorimetric))	-	-	Park et al., 2023 [96]
EBV	LAMP	*BALF5*	Serum	100 copies/tube	86.4%	100%	Iwata et al., 2006 [97]
HSV-1	LAMP	gG	Viral strain	500 copies/tube	-	99.9%	Pourhossein et al., 2011 [98]
HSV-1	LAMP	gG	Bronchoalveolar lavage fluid	NA	88% extract 80% boiling	95.7%	Vergara et al., 2019 [99]
HSV-1/2	LAMP	gG	Gingivostomatitis/ Vesicular Skin eruption patients	500/1000 copies (agarose) 1000/10,000 copies (turbidity)	-	-	Enomoto et al., 2005 [100]
HHV-6B	LAMP	*U31*	Serum	100 copies/reaction	94%	96%	Yoshikawa et al., 2014 [101]
HHV-6	LAMP	*U31*	PBMC	50 copies/tube	-	-	Ihira et al., 2004 [102]
HHV-6	LAMP	*U31*	Serum	10 copies/tube	95.5%	95.2%	Ihira et al., 2007 [103]
HHV-6A/B	LAMP	*U86*	Serum from HSCT patients	10 copies/tube (HHV-6A) 100 copies/tube (HHV6B)	-	-	Ihira et al., 2008 [104]
HHV-7		*U38*	Whole blood and plasma from febrile	500 copies/tube 30 min 250 copies/tube 60 min	-	-	Yoshikawa et al., 2004 [105]
HHV-8	LAMP	ORF26	Sarcoma tissue sample from KS/PEL	100 copies/tube	-	-	Kuhara et al., 2007 [106]
VZV	LAMP	ORF62	Swab samples from vesicular skin eruption patients	500 copies/tube	87%	100%	Okamoto et al., 2004 [107]
VZV		ORF62 (Oka vaccine) ORF62 (wild type)	Swab samples from varicella patients	100 copies/tube	-	-	Higashimoto et al., 2008 [108]
VZV		ORF62	Swab samples from skin eruptions (patients with varicella and vaccine) virus reactivation	5000 copies/µL	93.3% (varicella 84.4%	-	Higashimoto et al., 2019 [109]

Legend: CMV: Cytomegalovirus; LAMP: loop isothermal amplification mediated by loop; qLAMP: quantitative LAMP; EBV: Epstein–Barr virus; HSV-1: Herpes Simplex virus 1; HSV-2: Herpes Simplex Virus 2; HHV-6: Herpesvirus type 6; HHV-6B: Herpesvirus type 6B; HHV-6A: Herpesvirus type 6A; PBMC: Human Peripheral Blood Mononuclear Cells; HHV-7: Herpesvirus type 7; HHV-8; Herpesvirus type; HSCT: hematopoietic stem cells transplantation; KS/PEL: Kaposi’s sarcoma/primary effusion lymphoma.

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
