# Peer review of "Advancements in LAMP-Based Diagnostics: Emerging Techniques and Applications in Viral Detection with a Focus on Herpesviruses in Transplant Patient Management"

_ijms, 2024, doi:10.3390/ijms252111506_

Round 1

Reviewer 1 Report

Comments and Suggestions for Authors

This review article titled “Advancements in LAMP-Based Diagnostics: Emerging Techniques and Applications in Viral Detection with a Focus on Herpesviridae” authored by Ana Cláudia Martins Braga Gomes Torres, Carolina Mathias, Suelen Cristina Soares Baal, Ana Flávia Kohler, Mylena Lemes Cunha and Lucas Blanes highlights recent advancements in LAMP for virus diagnosis and explores current research trends and future prospects, emphasizing the detection challenges posed by Herpesviridae. The manuscript is well structured, comprehensive and covers all the important latest research articles. Furthermore, the topic is novel and important. The article is recommended for publication after addressing a couple of minor review comments:

1)    The language of the manuscript contains a few errors throughout the text which need to be rectified. The authors need to revise such minute errors throughout the manuscript before uploading.

2)    The authors are requested to discuss a few alternative NAATs in the Introduction section and justify the advantages of using LAMP over the other methods.

Comments on the Quality of English Language

The language of the manuscript contains a few errors throughout the text which need to be rectified. The authors need to revise such minute errors throughout the manuscript before uploading.

Reviewer 2 Report

Comments and Suggestions for Authors

The work presents quite an interesting overview of the LAMP technique and application from the last years. The reviewer suggests the publication in this journal.

Please, specify if any of the picture shown is reproduced from different works or not.

Author Response

Comments and Suggestions for Authors

The work presents quite an interesting overview of the LAMP technique and application from the last years. The reviewer suggests the publication in this journal.

Please, specify if any of the picture shown is reproduced from different works or not.

Response: Thank you for your comment. All figures in the manuscript were created by the authors and are original. None of the figures have been reproduced from previously published works.

Reviewer 3 Report

Comments and Suggestions for Authors

Summary of the Manuscript:

This review paper discusses the recent advancements in Loop-mediated Isothermal Amplification (LAMP) technology, highlighting its application in viral diagnostics with a focus on Herpesviridae. The article emphasizes the potential of LAMP, its modifications (such as DARQ-LAMP, QUASR, and FLOS-LAMP), and its advantages in point-of-care (POC) settings, especially for viruses like SARS-CoV-2. The paper also touches upon the challenges and future potential of LAMP in detecting herpesviruses, which have remained underexplored despite their clinical importance.

General Comments:

The paper is well-organized and provides an in-depth review of LAMP technology in viral diagnostics. The authors offer a comprehensive account of the technical developments in LAMP and their applications in detecting various viruses, including herpesviruses, which are of particular interest for immunocompromised patients. However, there are a few areas where the manuscript could be improved for better clarity, structure, and engagement with the audience.

Specific Comments:

  1. Introduction and Background:
    • The introduction is clear but could benefit from a more detailed explanation of why the detection of Herpesviridae using LAMP has been relatively underexplored. A stronger emphasis on the clinical significance of herpesviruses, especially in immunocompromised patients, would provide better context for the review's focus.
    • Consider expanding the section on the global burden of herpesvirus-related diseases to emphasize why improved diagnostics in this area are urgently needed.
  2. Methodological Clarity:
    • The section on LAMP methodology is technically accurate but somewhat dense. Some complex processes, such as the primer design and the amplification steps, could be broken down into simpler sub-sections to improve readability for a broader audience.
    • Including more visual aids, such as simplified diagrams or step-by-step flowcharts, might help readers better understand the LAMP process and its modifications.
  3. Tables and Figures:
    • The tables summarizing various studies on LAMP applications for viral detection are valuable, but the authors could enhance their discussion by providing more critical analysis. For example, explaining the limitations or success factors of LAMP in certain viruses versus others would give more context to the presented data.
    • In some cases, the limits of detection (LoD) and sensitivity data in the tables could benefit from additional commentary in the text. Discussing how these metrics compare across different studies or viral targets would make the data more meaningful.
  4. Discussion and Future Directions:
    • The manuscript would benefit from a more detailed discussion on the challenges associated with applying LAMP to Herpesviridae detection. For instance, discussing the technical difficulties or limitations in sensitivity when dealing with herpesviruses could provide insights into why this application has been slower to develop.
    • The authors should consider offering more concrete recommendations for future research. Specific areas where LAMP could be improved for herpesvirus detection, or suggestions for collaborative research efforts between molecular diagnostic labs and clinical institutions, could strengthen the paper's forward-looking perspective.
  5. Conclusion:
    • The conclusion effectively summarizes the potential of LAMP, but it could be enhanced by adding a more forward-looking statement. The authors should highlight specific ways in which LAMP can be expanded or improved for herpesvirus detection and its potential impact on clinical practices.
    • Mentioning the role of LAMP in the broader context of emerging molecular diagnostics and its potential integration with other technologies (e.g., CRISPR-based diagnostics) would make the conclusion more impactful.
  6. Language and Readability:
    • The manuscript is generally well-written but could be made more accessible by simplifying some of the technical jargon. This would help engage a broader readership, including clinicians or researchers from related fields who may not be as familiar with the intricacies of LAMP technology.
    • Definitions or a glossary for some of the more advanced LAMP modifications (e.g., DARQ-LAMP, FLOS-LAMP) would improve clarity for readers who may not be experts in molecular diagnostics.

Recommendations:

  • Major Revision Needed: While the manuscript is comprehensive and well-researched, some revisions are necessary to improve clarity and engagement, especially in the methodology section. More detailed explanations, simplified figures, and a stronger focus on the challenges and future directions would enhance the overall quality and impact of the paper.

Final Evaluation:

This manuscript addresses an important and timely topic in viral diagnostics, particularly in the context of emerging molecular technologies. With some adjustments, it has the potential to make a significant contribution to the field and inspire further research into the application of LAMP for the detection of herpesviruses. The review is informative and technically sound, but improving its accessibility and expanding certain discussions will make it a more valuable resource for a wider audience.

Comments on the Quality of English Language

Sufficient
